# Survey of Domestic Refrigerator Storage Temperatures in Poland for Use as a QMRA Tool for Exposure Assessment

**DOI:** 10.3390/ijerph20042924

**Published:** 2023-02-08

**Authors:** Constantine-Richard Stefanou, Anna Szosland-Fałtyn, Beata Bartodziejska

**Affiliations:** Institute of Agricultural and Food Biotechnology State Research Institute, 02-532 Warszawa, Poland

**Keywords:** refrigerator temperatures, domestic storage, exposure assessment, QMRA

## Abstract

In the framework of Quantitative Microbiological Risk Assessment, the estimation of the ingested dose of a hazard by the consumer is of paramount importance. This may be calculated by means of predictive modeling of growth/inactivation of the pathogen studied. For products that spend the majority of their shelf life in the domestic refrigerator, storage temperature will significantly impact the microbial population dynamics. To describe the variability of domestic storage temperatures in Poland, a survey including 77 participants, was carried out in Lodz, Poland. Participants were provided with temperature data loggers, which measured their refrigerator temperature for 24 h in 5-min intervals. The temperature-time profiles were used to calculate the mean working temperature, standard deviation, minimum and maximum values, and the data were statistically analyzed to find the best fitting probability distribution using R programming language. Out of the tested refrigerators, 49.35% had a mean working temperature of over 5 °C and 3.9% exceeded 10 °C. Distribution fitting scenarios were tested for goodness of fit, and the final selected distribution was a truncated normal distribution. This study can prove useful in Monte Carlo simulation analysis for stochastic quantitative food risk assessment in Poland.

## 1. Introduction

Perishable foods often require refrigeration to hinder microbial spoilage and/or pathogen growth. The controlled temperature of a refrigerator is meant to inhibit or delay the growth of microbes and particularly pathogens that could pose a threat to human health. Improper storing temperatures may allow for growth and increase the risk of foodborne disease. Domestic refrigerated storage is a crucial part of the cold chain, and proper storing temperatures contribute to food safety [1]. Control measures and good practice guides are in place for all other parties in the farm-to-fork path, including producers, processors, and distributors. Consumers, on the other hand, can be uninformed about proper food storage temperatures and practices [2]. Thus, temperature abuse or improper storage can occur during domestic refrigerated storage [3,4]. The importance of proper consumer storage and handling in terms of food safety is reflected in the reported source of outbreaks in the EU Zoonosis annual reports. For the period of 2016–2021, a total of 1411 household outbreaks occurred in Poland with the most commonly reported foodborne pathogen being *Salmonella* spp. [5]. A compounding factor for this is that most consumers have an optimistic bias when it comes to the domestic environment, viewing it as a safe place, hence, their risk perception at home is reduced. In addition to consumer behavior at home, the dietary shift towards ready-to-eat foods (RTE) draws attention to domestic storage as such products can be mildly processed with little or no preservatives and rely on respecting the cold chain to ensure safety and quality throughout the duration of their shelf-life [1].

For psychorotrophic bacteria in particular, temperature during storage can be the decisive factor for growth inhibition and control. Storage at increased temperatures could allow growth in contaminated products and lead to an increased risk of foodborne disease. Quantitative microbiological risk assessment (QMRA) requires the estimation of the ingested dose of a hazard by the consumer. This is calculated during the exposure assessment step by modeling the fate of the hazard in the food prior to consumption. Furthermore, reliable data on consumer storage are of paramount importance for establishing a shelf-life for perishable refrigerated foods [1].

Refrigerators can vary considerably, and the temperature fluctuates constantly during the cooling cycle. This variability in storage temperatures is of paramount importance, especially in stochastic Quantitative Microbiological Risk Assessment (QMRA) [6]. In deterministic QMRA, when simulating the consumer phase, the storage temperature is defined as a single point estimate, usually the mean or mode, instead of a probability distribution. This approach does not take into account the inherent variability of refrigerator working temperatures and may lead to erroneous final estimations. Therefore, the aim of this study was to collect data on domestic storage temperatures in Polish fridges and describe the variability of the storage temperature for use as input in stochastic risk assessment exercises and shelf-life determination.

## 2. Materials and Methods

Participants were provided with a temperature data logger (LogTag model TRIX-8, LogTag North America Inc., Auckland, New Zealand) and instructed to place it on the middle shelf of their refrigerator for 24 h. Participants were told not to change their refrigerator settings and to perform the measurement under normal usage conditions. This aimed at obtaining a realistic temperature data set that reflected everyday use and conditions. The survey carried out included 77 participants, inhabitants of Lodz in Poland, and took place from January to July of 2022. The data logger took temperature readings at 5-minute intervals, and these data points were used to calculate the mean temperature, standard deviation, minimum and maximum values for each refrigerator. 

The recorded mean temperature values were statistically analyzed to find the best-fitting probability distribution to describe the variability of Polish household refrigerator working temperatures. The probability distribution fitting was performed in R with the fitdistrplus package. Fitting was performed for the Weibull, triangular, lognormal, pert, and normal distributions, with parameters being estimated by using the maximum likelihood estimation (mle) method. All distribution fitting scenarios were statistically tested for goodness-of-fit. To select the distribution which best described the data, the Chi-squared, Kolmogorov–Smirnov (KS), Cramér–von Mises (CVM), and Anderson–Darling (AD) statistics were calculated alongside the Akaike’s Information (AIC), and Bayesian Information Criteria (BIC). These values were compared for all fitted distributions to confirm the best fit.

## 3. Results

### 3.1. Domestic Refrigerator Temperature Results

Indicative temperature profiles of Polish domestic refrigerators are presented in Figure 1. The data from all collected temperature profiles were used to calculate the mean working temperature of each individual refrigerator and construct a histogram for the distribution and the empirical cumulative density plot of the mean temperatures (Figure 2). A detailed summary of the recorded data is provided in Appendix A.

Table 1 presents the frequency of observed mean temperatures for the surveyed domestic refrigerators. Out of the tested refrigerators, 49.35% had a mean working temperature of over 5 °C and 3.9% exceeded 10 °C. Out of all the collected data points of all temperature profiles, 49.42% of the temperature readings were above 5 °C and 6.08% were above 10 °C. The mean working temperature was found to be 5.1 °C with a standard deviation of 2.8 °C, and minimum and maximum mean values were −1.2 °C and 15.2 °C, respectively.

A Box and Whisker plot was used to display the central tendency and dispersion of the mean working temperatures of the surveyed refrigerators (Figure 3). The 25th percentile (Q1) was 3.49 °C, and the 75th percentile (Q3) was 6.92 °C with an interquartile range (IQR) of 3.43 °C. The median was calculated as 5 °C and the ends of the whiskers were at −1.66 °C and 12.07 °C, respectively. The dot corresponds to the outlier value of 15.2 °C. 

The individual temperature profiles (Figure 1) displayed significant variation and hence, aiming to evaluate this variability for each profile, in addition to the mean working temperature, the standard deviation was also estimated from each temperature profile. The histogram of the calculated standard deviations for the 77 measured temperature profiles is shown in Figure 4. Relying on a single average value can prove inaccurate as the variability can range vastly from 0.3 °C to 3.7 °C, as seen in the figure.

### 3.2. Probability Distribution Fitting to Temperature Data

The fitting of probability distributions to the average temperature data was performed in R using the fitdist function. The data were fitted to the Weibull, triangular, lognormal, pert, and normal distributions. For each fitting, maximum likelihood estimation was used to estimate the probability distribution parameters. The goodness-of-fit statistics and criteria for each fitted distribution are presented in Table 2. Additionally, the bias factor for each fitted distribution was calculated using the Monte Carlo simulation method for 10,000 iterations, and the calculated mean bias factor is also presented in Table 2.

All the calculated statistics and criteria, with the exception of the Kolmogorov–Smirnov statistic pointed to the normal distribution as the best fitting distribution to describe the mean temperature of domestic refrigerators. The fitted normal would require truncation at the minimum and maximum recorded values of −1.2 °C and 15.2 °C. In contrast, the Kolmogorov–Smirnov statistic had the lowest value for the pert distribution. The suitability of the normal distribution as the preferred probability density function to describe these data was also evident in the histogram of theoretical densities and the P–P plot for all fitted probability distributions (Figure 5).

## 4. Discussion

Domestic storage is a key and often overlooked link in the food cold chain, and, to our knowledge, this survey is the first to focus on domestic storage temperatures in Poland. The survey conducted revealed that 88.31%, the majority of domestic refrigerators, function at over 4 °C, with an additional 3.9% having a mean temperature of above 10 °C. These numbers are in stark contrast with the European Food Safety Agency’s suggestion of below 5 °C for household refrigerators [7]. The suggested refrigerator temperature can vary among countries, but all agree on setting the limit at below 7 °C, with many countries lowering this to below 5 °C [8]. In our study, the temperature of the room in which the refrigerator was located was not considered, as Laguerre et al. noted no impact on the refrigerator temperature. Laguerre et al. also concluded that the temperature setting options of fridges did not influence the measured mean temperature and, as such, this factor was also excluded from the current study [9]. Drawing from the work of Jofré et al., door opening frequency was assumed to have little effect on the data [1]. The periodic fluctuations of temperature observed for each refrigerator, visible in Figure 1, are expected to be the result of the refrigeration cycle. Fluctuations during the course of the measurements for each refrigerator were observed, with the individual standard deviation estimated for each temperature profile ranging from 0.3 °C to 3.7 °C. James et al. note that temperature fluctuations in refrigerators can vary even for refrigerators with the same mean working temperature [8].

The surveyed households had fridges working at a mean temperature ranging from as low as −1.2 °C up to 15.2 °C. This noted variability in storage temperatures is of paramount importance for both food business operators (FBOs) and risk assessors aiming to guarantee the legally set appropriate level of protection (ALOP) for refrigerated foods. According to article 3 of Regulation (EC) No 2073/2005, FBOs are obliged to ensure that the food safety criteria applicable throughout the shelf-life of foods can be met under reasonably foreseeable conditions of distribution, storage, and use, including the consumer phase. Furthermore, the setting of use-by-dates for refrigerated pre-packed and ready-to-eat products requires applying stochastic methods for realistic estimations [10]. This is especially true in the case of psychotropic pathogens, such as *Listeria monocytogenes*, which are often detected in products meant for direct consumption [11]. 

The switch from single-point estimates for microbial growth parameters, such as storage temperature, to distributions requires a shift in perspective towards the full range of possible and probable values. Probability distributions reflect the variability of parameters, such as temperature, and aid in supporting more informed risk management decisions.

Based on similar studies in the literature, the Probability Density Function expected to best fit the data was expected to be the Normal distribution [1,10,12]. An adequate number of temperature data points was required for better distribution description and more precise fitting selection. For the statistical tests and criteria used to rank the fitted distributions, the lower the value the better fit of the distribution to the data. The Chi-squared, KS, and AD tests are often used to evaluate the acceptability of a theoretical distribution for the description of observed values [13]. The values of these statistics were used to compare the goodness-of-fit of several probability distributions to observed data. Additionally, the Akaike and Bayesian information criteria were used to estimate the prediction error and, therefore, relative quality of the fit. Applying these goodness-of-fit tests, the final domestic storage temperature distribution for Poland was described as a truncated normal distribution with a mean = 5.1 °C, sd = 2.8 °C and truncated at −1.2 °C and 15.2 °C. Roccato et al. in their analysis of domestic refrigerator temperatures in Europe, opted for describing the variability of European domestic refrigerators by a normal distribution with mean = 6.1 °C and sd = 2.8 °C for the northern European countries. Despite data from Poland not being included in the aforementioned study, our results are in relative agreement [10]. Future similar surveys could further validate our findings by including a larger number of participants and spanning a longer period of time. For all tested distributions, the bias factor estimated from a 10,000 iterations Monte Carlo simulation was within the acceptable range of 0.75–1.25.

## 5. Conclusions

The domestic environment can act as the setting of outbreaks of foodborne illnesses and storage temperature is a key factor for food safety at the consumer level, since it is a major controlling factor of microbial behavior in foods. A domestic refrigerator temperature survey was carried out for the first time in Poland, aiming to quantify the variability of mean storage temperatures. This study has provided a probability density distribution describing domestic refrigerator temperatures in Poland. The described truncated normal probability distribution (mean = 5.1 °C, sd = 2.8 °C, min = −1.2 °C, max = 15.2 °C) can be incorporated in Monte Carlo simulations for stochastic quantitative risk assessment and shelf-life determination. Efforts to quantify consumer phase storage variability increase the robustness and realism of risk assessments and, therefore, crucial risk management decisions critical for ensuring food safety.

## Figures and Tables

**Figure 1 ijerph-20-02924-f001:**
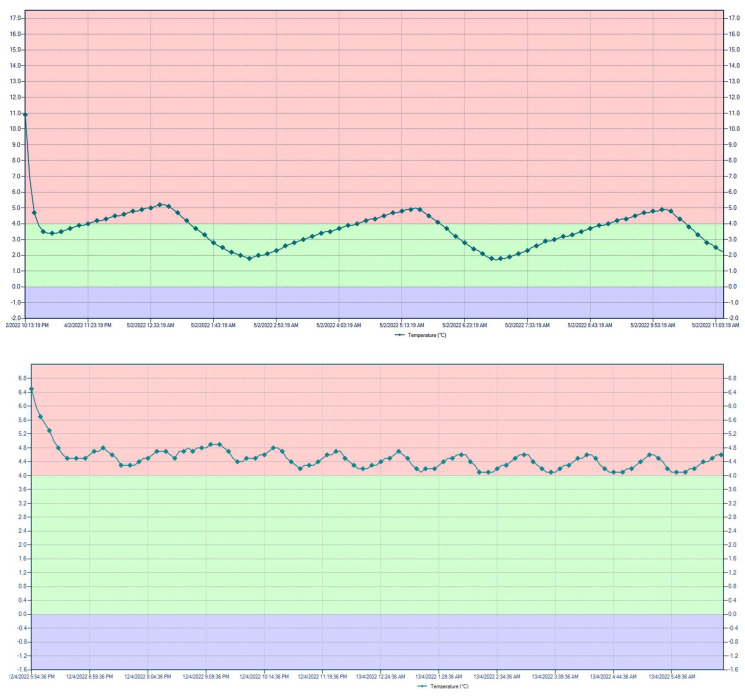
LogTag model TRIX-8 temperature profile of Polish domestic refrigerators.

**Figure 2 ijerph-20-02924-f002:**
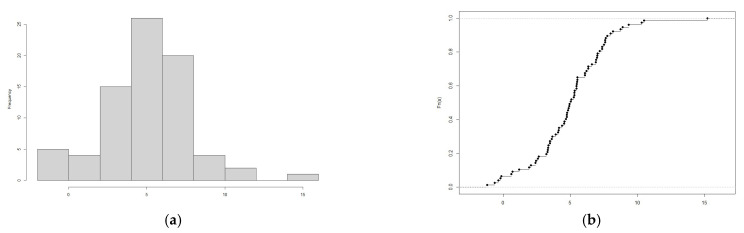
(**a**) Distribution of the mean working temperatures of Polish household refrigerators, (**b**) Empirical Cumulative Density function of the mean working temperatures of Polish household refrigerators.

**Figure 3 ijerph-20-02924-f003:**
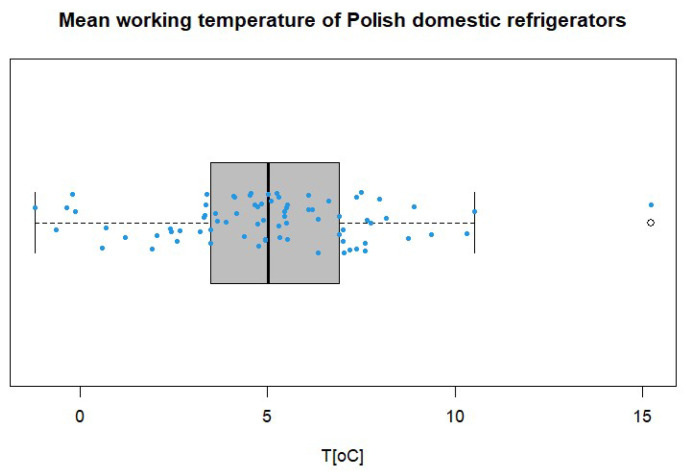
Box and whiskers plot of domestic storage temperature data.

**Figure 4 ijerph-20-02924-f004:**
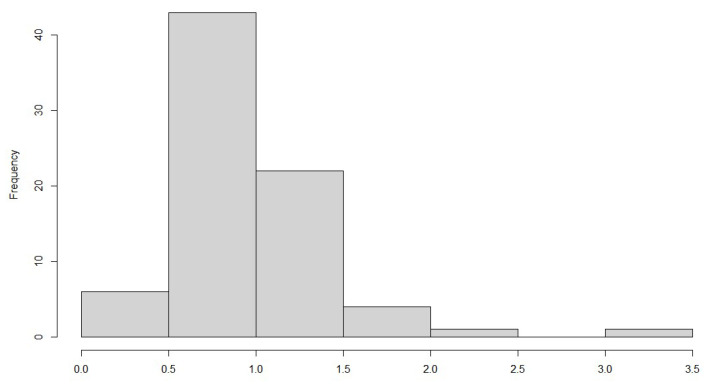
Histogram of the standard deviation of recorded temperature profiles.

**Figure 5 ijerph-20-02924-f005:**
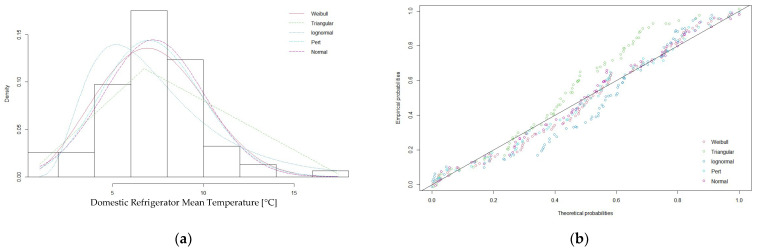
(**a**) Histogram of theoretical densities and (**b**) P–P plot for the fitted probability distributions.

**Table 1 ijerph-20-02924-t001:** Frequency table of observed mean temperatures in Polish household refrigerators.

Mean Temperature [°C]	T < 0	0 ≤ T < 2	2 ≤ T < 4	4 ≤ T < 6	6 ≤ T < 8	8 ≤ T < 10	10 ≤ T < 12	T > 12
**Frequency**	5	4	15	26	20	4	2	1
**Percentiles**	6.49%	5.19%	19.48%	33.77%	25.97%	5.19%	2.60%	1.30%

**Table 2 ijerph-20-02924-t002:** Calculated goodness-of-fit statistics and criteria used for best fit selection.

Goodness-of-Fit Statistics	Weibull Distribution	Triangular Distribution	Lognormal Distribution	Pert Distribution	Normal Distribution
Chi-squared	15.6	30.5	31.4	13.9	12.9
Kolmogorov–Smirnov	0.01	0.21	0.16	0.08	0.08
Cramer–von Mises	0.14	0.82	0.55	0.09	0.08
Anderson–Darling	0.87	4.41	3.23	0.60	0.51
**Goodness of fit criteria**					
AIC	379.99	396.28	402.76	382.46	378.88
BIC	384.68	403.30	407.45	391.83	383.57
Bias Factor (Bf)					
Mean Bf	1.0010	1.1071	1.0098	1.0006	1.0093

## Data Availability

Not applicable.

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
