# Peer review of "Survey of Domestic Refrigerator Storage Temperatures in Poland for Use as a QMRA Tool for Exposure Assessment"

_ijerph, 2023, doi:10.3390/ijerph20042924_

Round 1
Reviewer 1 Report
In this article, the authors conducted a survey of domestic refrigerator storage temperatures in Poland. The topic has certain application value, and has certain guiding significance for risk assessment. However, the data is relatively simple and is not enough to support the proposed conclusions. Moreover, the experimental design needs to be improved by including more participants and extending the survey periods. Detailed comments are as below.
(1) The article type can be changed to Communication due to the limited data presented.
(2) The Abstract was not well written. The introduction part was too long, while the results part was too short.
(3) “Food safety” and “Poland” in the Keywords should be removed.
(4) Lines 41-42: Why not discuss the survival of Salmonella spp. under the observed temperatures in this study?
(5) Line 58: Please spell out QMRA.
(6) Lines 71-72: Please include more participants and extend the survey periods in order to draw a solid conclusion.
(7) Line 87: Please divide the Results section into several parts and add some subtitles.
(8) Line 150: The Discussion part was not well constructed. It should be focused on the major findings in your work.
(9) Line 207-208: Please indicate the implications of this finding for food safety.
(10) How will this work contribute to QMRA?
Author Response
Response to Reviewer 1 Comments
Please also find the revised manuscript attached below
Point 1: The article type can be changed to Communication due to the limited data presented.
Response 1: Thank you for this suggestion. We have considered this option but believe that our manuscript could be submitted as an article. Additionally, reviewer 2 did not suggest changing the type of the manuscript.
Point 2: The Abstract was not well written. The introduction part was too long, while the results part was too short.
Response 2: Thank you for your suggestion. In order to balance out the abstract we have reduced the introduction part.
Point 3: “Food safety” and “Poland” in the Keywords should be removed.
Response 3: Thank you for your suggestion. These keywords have been removed in the revised manuscript (line 25).
Point 4: Lines 41-42: Why not discuss the survival of Salmonella spp. under the observed temperatures in this study?
Response 4: Thank for the question we have considered your suggestion. We did wish to focus too much on specific organisms, aiming to present the probability distribution describing temperature variability as the main focus.
Point 5: Line 58: Please spell out QMRA.
Response 5: Thank you for the correction, this flaw has been corrected in the manuscript (line 59)
Point 6: Lines 71-72: Please include more participants and extend the survey periods in order to draw a solid conclusion.
Response 6: We agree with your point and ideally more participants would be included, but unfortunately, it's beyond the scope of the current study to explore that. In future studies we recommend collecting larger data sets in order to cross validate or confirm our findings and extend the data collection period. Until that time our findings should be considered preliminary hence, we've acknowledged this in the Discussion (Line 209)
: “Future similar surveys could further validate our findings by including a larger number of participants and spanning a longer period of time.”
Point 7: Line 87: Please divide the Results section into several parts and add some subtitles.
Response 7: Thank you for your suggestion, this would improve the readability and structure. We have divided the results into two sections:
3.1 Domestic refrigerator temperature results (line 89)
3.2 Probability distribution fitting to temperature data (line 133)
Point 8: Line 150: The Discussion part was not well constructed. It should be focused on the major findings in your work.
Response 8: Thank you for the constructive criticism. We have altered the discussion and have implemented suggestions to improve the section of the manuscript.
Point 9: Line 207-208: Please indicate the implications of this finding for food safety.
Response 9: We appreciate your point and have altered the conclusions section to highlight the importance of variability incorporation into risk assessment for food safety (line 215-225).
Point 10: How will this work contribute to QMRA?
Response 10: The application of QMRA in a country specific case requires temperature data for use in exposure assessment at the consumer phase. As such data was lacking for domestic storage in Poland, we believe that our findings will aid risk assessors and regulatory authorities in Poland and the EU in assessing more realistically the potential growth/survival of pathogens improving the QMRA.

Reviewer 2 Report
Dear editor,
The submitted manuscript reports storage temperature data and its fitting to statistical distributions, which are essential parts of many quantitative microbial risk assessments. Therefore, I support acceptance of the manuscript after minor revisions.
If there are no ethical consequences, I strongly suggest that the raw data should be made available. Risk assessors would most likely use the data for fitting or combining with other studies. The fluctuations in the temperature are also quite important for dynamic predictive microbiology studies, so they can extract valuable information from this study.
Title: Abbreviation should not be used in the Title.
Keyword: Please consider revising keywords. Keywords that already appear in the title will reduce the chances of getting found from a database search.
Line 42: Salmonella should be italicized
Line 46: I believe most RTE foods are quite heavily processed. However, post-process (or post-lethality) contamination during packaging is an important concern, especially for Listeria monocytogenes. After an RTE food is contaminated with Listeria, it can grow in the refrigerator, as it is a psychrotroph.
Line 67: Data loggers were provided to the participants and not placed by the researchers, do you anticipate any bias resulting from user error?
Line 79: So was mean temperature for each refrigerator used for fitting? That would underestimate the variance as each refrigerator also has fluctuations, if the error was not propagated.
Could you please update Figure 1? Too small. Red dot (Inspection) in the legend does not appear on the actual plot, therefore, should be removed.
Line 105: It may also be important to know that how often the fluctuations resulted in temperatures over 5 °C and 10 °C.
Table 1: Please consider adding percentiles in addition to frequency.
Figure 3: Can you add the data points to the boxplot?
Lines 120-126: From Figure 1, it can be seen that the temperature fluctuations are rather periodic. I think that the fluctuations in temperature were not discussed enough throughout the manuscript. Can the authors consider giving a few more distinct examples given in Figure 1 to better explain if there are any specific differences between different refrigerators? This is particularly important for predictive microbiology studies with dynamic temperature conditions.
Table 2: Too many digits.
Lines 181-182: This sentence is not very clear. Does it mean that you expected that the data would fit to Normal?
Lines 181-197: Cumulative distribution functions can also be used for QMRA. For example, @Risk has a function called “@RiskCumul” to directly insert probability and data points at the corresponding probability values. Your Table 1 can be directly used as an input, without any distribution fitting.
I suggest calculating the bias factor for the fitted distributions (probably with a Monte-Carlo simulation). For example, from the histogram, using the lognormal distribution will result in lower temperatures than the actual data, causing a fail-dangerous situation for the exposure assessments based on the data. Providing bias factors will help to define the direction and magnitude of such differences.
Author Response
Response to Reviewer 2 Comments
Please also see the attached revised manuscript.
Thank you for your comments and suggestions and taking the time to review our work.
Our response to the points raised are as follow:
Point 1: If there are no ethical consequences, I strongly suggest that the raw data should be made available. Risk assessors would most likely use the data for fitting or combining with other studies. The fluctuations in the temperature are also quite important for dynamic predictive microbiology studies, so they can extract valuable information from this study.
Response 1: Thank you for the suggestion. We agree that such data would be very useful for felow researchers and following your comment, we have added a table with the recorded mean temperatures, standard deviation, min, max and range values to Appendix A as “Table A1. Raw data of the surveyed domestic refrigerators.” (Line 237). In regards to dynamic profiles, the individual temperature-time profiles unfortunately are too cumbersome to include in a manageable format, and as such we think it best to include the statistics of each individual surveyed refrigerator.
Point 2: Title: Abbreviation should not be used in the Title.
Response 2: We agree that this is not ideal but the title would be quite long if we wrote it out fully. Having searched through the literature some articles have included QMRA in the title, but I understand this is not a good practice.
Point 3: Keyword: Please consider revising keywords. Keywords that already appear in the title will reduce the chances of getting found from a database search.
Response 3: Thank you for your comment, we have considered the keywords and removed 2 as also requested by Reviewer 1. We believe they are appropriate for the topic.
Point 4: Line 42: Salmonella should be italicized
Response 4: Thank you for spotting that error, we have corrected the mistake in the manuscript.
Point 5: Line 46: I believe most RTE foods are quite heavily processed. However, post-process (or post-lethality) contamination during packaging is an important concern, especially for Listeria monocytogenes. After an RTE food is contaminated with Listeria, it can grow in the refrigerator, as it is a psychrotroph.
Response 5: Thank you for the insightful point. In this case we were referring to products such as prepacked salads, sliced fruits etc which literature shows are on the rise, but we agree that the wording is not clear and should be corrected.
We have altered that part of the manuscript (line 45) so it does not sound as if all RTE are mildly processed. The text now reads:
“In addition to consumer behaviour at home, the dietary shift towards ready to eat foods (RTE) draws attention to domestic storage as such products can be mildly processed with little or no preservatives and rely on respecting the cold chain to ensure safety and quality throughout the duration of their shelf-life [1].”
Point 6: Line 67: Data loggers were provided to the participants and not placed by the researchers, do you anticipate any bias resulting from user error?
Response 6: Thank you for your comment. Participants were instructed in the correct use of the data loggers individually before receiving them, being shown the correct use and did not include any bias resulting from user error.
Point 7: Line 79: So was mean temperature for each refrigerator used for fitting? That would underestimate the variance as each refrigerator also has fluctuations, if the error was not propagated.
Response 7: Thank you for your comment and suggestion. To our knowledge this method has been followed by other researchers when describing the distribution of domestic storage temperatures (Koutsoumanis et al.). We did not consider the variance for each refrigerator independently but took into account the standard deviation for this reason. We believe that the standard deviation can be used to describe this fluctuation during the refrigeration cycle but future surveys could include both statistics to provide a cleared picture.
Point 8: Could you please update Figure 1? Too small. Red dot (Inspection) in the legend does not appear on the actual plot, therefore, should be removed.
Response 8: Thank you for your comment. Figure 1 has been augmented to improve clarity but formatting the plots is not convenient due to the specific software of the LogTags used, which restricts the editing options for the plots.
Point 9: Line 105: It may also be important to know that how often the fluctuations resulted in temperatures over 5 °C and 10 °C.
Response 9: Thank you for your suggestion. We agree that this would be interesting and have added the percentage of reading above these 2 temperatures in the result section.
The text now reads: “Out of all the collected data points of all temperature profiles, 49.42% of the temperature readings were above 5 °C and 6.08% were above 10 °C.” (Line 108)
Point 10: Table 1: Please consider adding percentiles in addition to frequency.
Response 10: Thank you for the suggestion. We agree that such information would make further use of our data easier for other researchers and as such have included the percentiles in Table 1.
Point 11: Figure 3: Can you add the data points to the boxplot?
Response 11: We appreciate this would improve the visualization and the plot has been updated in the manuscript to include the data points as suggested.
Point 12: Lines 120-126: From Figure 1, it can be seen that the temperature fluctuations are rather periodic. I think that the fluctuations in temperature were not discussed enough throughout the manuscript. Can the authors consider giving a few more distinct examples given in Figure 1 to better explain if there are any specific differences between different refrigerators? This is particularly important for predictive microbiology studies with dynamic temperature conditions.
Response 12: Thank you for your point. We agree that this is lacking as the presentation of the distribution of the individual profile standard deviations does not convey the full picture. An additional second temperature profile has been added to Figure 1 to show this periodic fluctuation between refrigerators. We have also added to the discussion, addressing the differences between refrigerators and reinstating the range of estimated standard deviation values.
The text now reads: “The periodic fluctuations of temperature observed for each refrigerator, visible in Figure 1, are expected to be the result of the refrigeration cycle. Fluctuations during the course of the measurements for each refrigerator were observed, with the individual standard deviation estimated for each temperature profile ranging from 0.3 °C to 3.7 °C. James et al. note that temperature fluctuations in refrigerators can vary even for refrigerators with the same mean working temperature [8].” (Line 170)
Point 13: Table 2: Too many digits.
Response 13: Thank you for the correction. The values in the table have been changed
Point 14: Lines 181-182: This sentence is not very clear. Does it mean that you expected that the data would fit to Normal?
Response 14: Based upon similar studies in the literature, the vast majority of surveys concluded that storage temperature was best described by a normal distribution, hence our expectation that our data would also quite likely fit a normal distribution. Since this sentence was not very clear we have changed it in the manuscript to: “Based on similar studies in the literature, the Probability Density Function expected to best fit the data was expected to be the Normal distribution.” (Line194)
Point 15: Lines 181-197: Cumulative distribution functions can also be used for QMRA. For example, @Risk has a function called “@RiskCumul” to directly insert probability and data points at the corresponding probability values. Your Table 1 can be directly used as an input, without any distribution fitting.
Response 15: We agree with your point and have included the percentiles in Table 1 as suggested so this is accessible to readers. There is also a plot of the empirical cumulative distribution in Figure 2.b (Line 102) and the quantile values are given below the boxplot, Figure 3, allowing for the cumulative distribution to be defined by fellow researchers.
Point 16: I suggest calculating the bias factor for the fitted distributions (probably with a Monte-Carlo simulation). For example, from the histogram, using the lognormal distribution will result in lower temperatures than the actual data, causing a fail-dangerous situation for the exposure assessments based on the data. Providing bias factors will help to define the direction and magnitude of such differences.
Response 16: We agree that including the Bias factor would be of interest. These have been calculated for 10,000 iterations of Monte Carlo and the mean Bf have been included in Table 1. and mentioned in the text (Line 138 & 209)

Round 2
Reviewer 1 Report
The manuscript is acceptable in its current form.